# Can Plants Sense Humans? Using Plants as Biosensors to Detect the Presence of Eurythmic Gestures

**DOI:** 10.3390/s23156971

**Published:** 2023-08-05

**Authors:** Luis de la Cal, Peter A. Gloor, Moritz Weinbeer

**Affiliations:** 1MIT Center for Collective Intelligence, Cambridge, MA 02142, USA; l.delacal@alumnos.upm.es; 2Escuela Técnica Superior de Ingeniería de Sistemas Informáticos, UPM Technical University of Madrid, 28031 Madrid, Spain; 3Foundation Fintan, 8462 Rheinau, Switzerland; moritz.weinbeer@bdas.ch

**Keywords:** biosensors, machine learning, motion detection, plant–human interaction

## Abstract

This paper describes the preliminary results of measuring the impact of human body movements on plants. The scope of this project is to investigate if a plant perceives human activity in its vicinity. In particular, we analyze the influence of eurythmic gestures of human actors on lettuce and beans. In an eight-week experiment, we exposed rows of lettuce and beans to weekly eurythmic movements (similar to Qi Gong) of a eurythmist, while at the same time measuring changes in voltage between the roots and leaves of lettuce and beans using the plant spikerbox. We compared this experimental group of vegetables to a control group of vegetables whose voltage differential was also measured while not being exposed to eurythmy. We placed a plant spikerbox connected to lettuce or beans in the vegetable plot while the eurythmist was performing their gestures about 2 m away; a second spikerbox was connected to a control plant 20 m away. Using *t*-tests, we found a clear difference between the experimental and the control group, which was also verified with a machine learning model. In other words, the vegetables showed a noticeably different pattern in electric potentials in response to eurythmic gestures.

## 1. Introduction

The notion of plant perception has been a subject of great interest among scientists for several decades, and recent research has shown that plants are capable of sensing and responding to their environment in ways previously unknown [1,2,3]. One of the most fascinating aspects of plant behavior is their capacity to perceive and respond to external stimuli, such as light, sound, and touch. However, the extent to which plants can detect and respond to the movements of nearby organisms, particularly humans, is not yet clear. Eurythmy is a form of movement that is believed to have a harmonizing effect on the body and mind [4]. It involves movements and gestures of the arms, legs, and torso, often accompanied by music or speech. The hypothesis of this study is based on the idea that the movements generated by eurythmy may have an impact on the electrical activity of nearby plants, which can be detected using specialized equipment.

To test this hypothesis, we designed an experiment to measure electric differences of plants [5], while an actor performed eurythmic gestures in their vicinity. In particular, an actor performed eurythmy on a garden plot near rows of lettuce and beans, while a highly sensitive voltage meter measured differences in voltage within a plant near and exposed to the eurythmist. At the same time, a second similar measurement device was connected to a control plant further away. We hypothesized that the electric activity of the plants would change in response to the movement generated by the eurythmist, and that this change would be detectable using electrical sensors. Our method is inspired by the results that famous Indian physicist Jagadish Chandra Bose obtained more than a hundred years ago [6]. He measured the electric reactions of *Mimosa pudica* and *Codariocalyx motorius* to external triggers using sophisticated measurement equipment he constructed himself.

The purpose of this study was to determine the extent to which the hypothesis holds true, and to explore the potential implications of our findings for our understanding of the relationship between humans and plants. By investigating the impact of eurythmy on the electrical activity of plants, we hope to shed new light on the ways in which plants interact with their environment and respond to external stimuli. Ultimately, our goal is to deepen our understanding of plant awareness, and to explore the many possibilities that arise from this understanding.

## 2. Related Work

The field of using plants as biosensors has an extensive history applied to various different fields. The pioneer in this field is Indian scientist Jagadish Chandra Bose, who already at the end of the 19th century showed that plants exhibit electrical responses to external stimuli [6]. Bose invented their own ingenious measurement instruments to demonstrate electric communication within plants, analogous to the nervous systems of animals [1]. In newer research, electrical signals of plants have been used as biosensors to measure the direction of light, monitor the environment, and detect insect attacks and the effects of pesticides and pollution [7]. Researchers from the same team also found that electrostimulation of tomato plants was picked up by neighboring tomato plants [3]. More recently, it has even been shown that plants also communicate by sound [2]. Besides using plants as sensors by measuring their electrical voltage differentials, Kovalchuk et al., 2008 [8], in the field of environmental pollution detection, proposed the use of transgenic plants to monitor the presence of genotoxic pollutants in the air.

Different studies have investigated the effect of eurythmy on plants, most covering basic research. The work of the Institute ArteNova (Basel) with Tanja Baumgartner, Eckart Grundmann, and others is particularly noteworthy. To name a few examples, for instance, taste development in apples [9], shape formation in different plant species [10], and changes in heredity [11] were studied, where plants were put under the influence of eurythmy. Other researchers looked at the influence of different eurythmy gestures on the overall shape formation of lettuce [12] or *Bryophyllum* [13]. In a two-year project, the effect of two different eurythmy gestures on a total of six different plant species was investigated [14].

In these studies, depending on the methodology, in some cases a small and in others a significant influence of eurythmy on the analyzed properties of the plants was observed. Furthermore, depending on the gesture, eurythmy had an inhibiting or promoting effect on gestalt formation and growth processes. Applied criteria for measuring the influence were both qualitative and quantitative, while some of the studies mentioned leaving room for interpretation in their scientific design, others followed strictly scientific criteria.

However, recent advances are more scarce when looking at the application of plants as human biosensors. A recent example in the field is work by Oezkaya et al., 2020 [15], where the authors achieved 66% accuracy in identifying individuals based on their walking patterns, captured by the potential difference in a plant using a Plant Spikerbox (https://backyardbrains.com/products/plantspikerbox (accessed on 1 May 2023)). The work also achieved an 85% accuracy when classifying whether the walk represented a positive or a negative mood. A main takeaway of this work is that there is potential in using plants as biosensors, and they can offer more information than expected at first. In their experimental process, they used MFCC (Mel-frequency cepstral coefficients) features from the Mel-Frequency-Spectogram that were obtained from Spikerecorder software. MFCCs are the coefficients that together describe the shape of the spectral envelope and are frequently used for voice processing [16]. These features were then processed by a Random Forest tree classifier to generate predictions on a test set. Similar to this work, our study aims to expand the field of using plants as biosensors using machine learning, but taking into account different sound features from the plant recordings and contrasting the results of different machine learning techniques.

One question which this research does not address is how the plants capture the human movement. In fascinating research, Abe et al., 2014 [17] constructed a visual microphone that was able to reconstruct sound using a high-frequency camera that captured the tiny oscillation of objects such as an empty bag of chips when they were hit with sound waves. We can speculate that a similar effect might be at play here, where the oscillations of the human moving near the plant have small effects on the plant.

Similarly to the previously mentioned study, the work of Duerr et al., 2020 [18], studied voltage differentials in plants during human interactions. In this study, the researchers measured the potential variations of plants and recorded the leaf movement to correlate these measurements with the movements of a eurythmist. The results show some correlation between these factors, and even though the study has limitations because of the small sample size, it opens the door to further research in this field. Our study aims to expand this experiment by having a bigger sample size and applying machine learning techniques to classify whether eurythmic gestures are currently being performed or not.

## 3. Hypothesis

Based on the previous research laid out in the related work section, we speculate that a plant will somehow sense its surroundings and respond accordingly. The purpose of this research is to show that a plant is highly reactive to perceived changes in its environment. Note that we do not make any claims about how this mechanism works.

We hypothesize that a plant will exhibit differences in voltage, measured by electrodes connected near its root and at its leaves, in response to changes in its environment. In particular, we conjecture that through the voltage difference recorded from different plants using specialized equipment, one can distinguish whether someone is performing eurythmic gestures in the vicinity of the plant or not. The purpose of this study was to determine whether the hypothesis holds true and to what extent this difference is perceivable.

## 4. Method

Figure 1 shows the setting where the data collection was performed, namely an agricultural field in the north of Switzerland. In this area, different plants were selected and a voltage measurement device called Plant Spikerbox was connected to them. This tool has two electrodes that need to be connected to the ground and to the leaf of the plant, respectively. Using these connection points, it tracks the potential differences of the plant in real time. To do this, it uses Spikerecorder Software, that converts these signals into wav sound files that can later be processed in other programs.

The recordings were performed taking into account similar weather patterns across the experiment to reduce the noise inherent in the experimental setting. The eurythmist stood at the border of a plot of garden plants, about 2 m away from the plant to which the spikerecorder was connected (Figure 2). At the same time, a control reading was taken from another plant connected to another spike recorder, positioned about 20 m away from the eurythmist and growing under similar, comparable environmental conditions (soil, humidity, light, wind, etc.) to the plant exposed to eurythmy. It might be that the distance of the eurythmist to the plant could influence the strength of the reading; however, this was not part of the experiment. Instead, we tried to keep the distance constant for all the readings.

Multiple recordings were taken from different plants, including lettuce plants and beans, to create a more diverse dataset consisting of different plant species. During the experiment, six recordings were taken from the same plant on a weekly basis. At the same time, “control readings” were performed simultaneously by collecting spikerbox recordings of plants without any exposure to eurythmy. On the other hand, recordings tagged as “eurythmy” were taken while an actor was performing eurythmic gestures next to the plant. In this way, we were able to capture the voltage differentials of “experimental” and “control” plants, allowing us to analyze a dataset with two populations of vegetables, trying to find distinctive differences in patterns and build classification machine learning models to distinguish between beans and lettuce plants exposed to eurythmy or not.

After data cleaning because of malfunctioning spikerboxes, the final sample contained 28 control files and 22 eurythmic files. The process applied in the experiment is shown in Figure 3. After the voltage changes in the form of wav files were recorded with the SpikeRecorder, the files were processed using pyAudioAnalysis [19]. This tool extracts the audio features that can later be used to train different machine learning classification models.

Regarding machine learning classification models, we tested both classical approaches and deep learning models. Firstly, we tested traditional machine learning models from the scikit learn library [20]. The classifiers that we took into account in our experimental phase are the following.

Gaussian Naive Bayes.Random Forest with 100 estimators.Extra Trees with 500 estimators.Support Vector Machine with linear kernel and 1.0 regularization.Gradient Boosting with 500 estimators.

Secondly, we used Tensorflow [21] to create a fully connected deep neural network consisting of multiple dense layers with rectified linear unit (ReLU) activation functions and dropout regularization. The first layer is a dense layer which serves as the input layer; the maximum norm of weights was also limited to encourage robust generalization. Next, a dropout layer was added with a 0.2 rate, which randomly sets 20% of the inputs to 0, thus reducing overfitting by reducing the interdependencies between layers. Next, a new dense layer is added, which is identical to the first one, except that the input dimension is reduced by half. Next, a new dropout layer is added to further improve the model’s robustness. Lastly, an output dense layer is added with a sigmoid activation, which is common in binary classification problems, to produce probabilities between 0 and 1.

## 5. Results

Figure 4 illustrates an example of the potential variations between a sample plant treated with eurythmy on the left and an untreated sample plant on the right. As Figure 4 indicates, a plant exposed to eurythmy shows a much stronger potential signal with a consistently higher voltage. Note that the values on the y-axis are relative; the spikerbox software makes no claims about the absolute value of the amplitude.

Table 1 describes the features which have been computed with pyAudioAnalysis which are significantly different between the eurythmy and the control plants.

In Table 2, we can observe the results with the lowest *p*-value after applying a *t*-test to all of the columns of the two different data classes (Control and Eurythmic). If we set the significance level to 0.005, features with a *p*-value lower than this will indicate that the difference between the two classes is unlikely to have occurred naturally by mere chance. We found that 71 features have a *p*-value below the 0.005 threshold and therefore there are many characteristics that have a statistically significant difference between the two classes. The Cohen’s d measurement that is displayed for each variable also suggests a moderate effect of the variables on the classification, indicating a meaningful difference between the control and eurythmy groups [22].

Once we have determined that the two different classes have statistically significant differences, we can try training machine learning models to classify an audio file as either Control or Eurythmic.

Using the same training and test split for every experiment, we created popular classical machine learning classifiers. The specific parameters used in their creation can be found in Section 4. For each trained classifier, we used the model to predict the values of the test dataset, obtaining their f1-score, accuracy, precision, and recall values, which can be found in Table 3. The confusion matrices for all classification models can be found in Figure 5. It is important to note that apart from Naive Bayes, Random Forest, Extra Trees, Support Vector Machine, and Gradient Boosting classifiers, we have included the results for the majority class as a baseline, which involves classifying all of the data as the majority class, in this case the control. This can aid by putting the results into perspective. To aid in analysis of this table, the best results for each metric have been highlighted. When looking at the model with highest recall and f1-score, which is Naive Bayes, we could initially think that the model has performed well, but taking a closer look at the metrics, we can see that it has an even lower accuracy than the baseline, predicting correctly only half of the time. Its confusion matrix shown in Figure 5b reveals that the recall is so high because most of the values have been classified as Eurythmic, causing most of the true eurythmic values to be classified correctly. However, the high number of false positives makes this a low-quality model. Regarding the model with the best accuracy, which is Random Forest, we see some promising results. In Figure 5c, we can see that even though the number of false negatives is slightly higher than true positives, which impacts the recall negatively, the precision is higher than average, which means that when our model makes a eurythmic prediction, it has a higher chance of being right. However, the model with the best precision in our experiments is SVM, which when paired with the highest true negative rate of the explored model, makes it an interesting option. In conclusion, all models except for Naive Bayes show similar accuracy and precision values and some promise in the accurate detection of eurythmic movements for our specific tests, though the f1-score and recall values show some room for improvement.

Now that we have seen the results of the traditional machine learning methods, we will test our dataset against a fully connected deep neural network. The composition of the different layers that make up this model is described in Section 4. Even though deep learning models tend to benefit from larger quantities of data, we think that it is important to evaluate and compare them to traditional machine learning methods in this scenario because of their remarkable success in various domains.

Several experiments were performed, creating a range of options by varying the number of epochs and the validation split. The best results for the classification problem were found by restoring the best weights regarding the minimum validation loss with 100 epochs and a validation split of 0.2. We achieved an accuracy of 62% and an f1-score of 65%, as well as a precision of 58% and a recall of 72% in the classification task. Even though the results in Figure 6 show a relatively large number of false positives (control files classified as eurythmy), the low number of false negatives shows that when our model classifies a file as control, it has a higher chance of being correct, resulting in a recall of 72%. The highest values in the confusion matrix are also correctly identified control values and correctly identified eurythmic values. Therefore, the model shows some promise in distinguishing between plants in their normal state and plants which are experiencing eurythmic gestures being performed next to them.

Figure 7 depicts the computed SHAP (Shapley additive explanations) values for the generated deep learning model. These SHAP values provide insights into the individual feature contributions to the model’s output [23]. The analysis reveals that three specific features, namely *mfcc_1_std*, *mfcc_1_mean*, and *delta mfcc_1_std*, exhibit a consistently high average impact on the model’s predictions.

Taking into consideration the SHAP values, we created a subset of the training and testing dataset, selecting only the 10 features with the highest average SHAP values in the previous best model. This was done to test the metrics of deep learning models created using only these features. We again created a set of variations with a range of epochs and validation splits. The best performing model was achieved by restoring the best weights regarding the minimum validation loss with parameters of 200 epochs and a validation split of 0.25. We achieved an accuracy of 63% and an f1-score of 61%, as well as a precision of 61% and a recall of 60% in the classification task. The results of the predictions can also be seen in the confusion matrix in Figure 8. When comparing these results with the ones using all of the features, we can see that there was almost no negative impact on f1-score and the accuracy even slightly increased. Even though the recall is lower, the precision slightly increased, which means that the model correctly predicted eurythmic values more accurately. All of this was achieved with a feature reduction of more than 735%, going from 136 original features to just 10.

## 6. Discussion

In this experiment, we have investigated the hypothesis that plants will respond differently when exposed to human body movements. In particular, a eurythmist conducted movements and gestures while focused on lettuce or beans two meters away. Using t-tests, we were able to predict a significant difference in the potential readings of experimental plants compared to control plants. Using machine learning, we were also able to predict if a particular reading of a one to two minute voltage reading sample was from a plant exposed to eurythmy, or from a control plant five to six meters away and not exposed to eurythmy, with 63 percent accuracy. While this indicates a relationship between the eurythmic gestures of the actor, it is not clear how precisely the plant picks up the movement of the human. Towards this goal, much further work is required.

However, at the very least, these results are an additional indication that a plant might be aware of its environment. As has already been mentioned in the related work section, plants are aware of sounds, movement, and changes in electrical voltage applied to the plant. In this research, we find that plants might also be sensitive to rhythmic human body movements.

While this work is very preliminary and experimental, it carries high theoretical and practical value. Although in this research we focus on the plants’ perceptions of eurythmic gestures, these insights can easily be generalized. On the theoretical side, demonstrating that plants perceive human movement and developing and testing a method that can capture that perception of human movement of the plant will enable new research on the possible sentience of plants. It will thus continue the work conducted at the beginning of the 20th century by Jagadish Chandra Bose, who said that plants have life and “that everything in man has been foreshadowed in the plant”.

On the practical side, this research opens the door for using plants as biosensors of human movement. For instance, plants could be used as sensors for capturing and categorizing different types of human movement, measuring human emotions [15], or for using plants as privacy-respecting sensors for distinguishing between different people [15].

## 7. Limitations

While the results we observed are encouraging, this study has many limitations. The first one is the question of causality, as we can only claim correlation, but not causation. Although we found changes in voltage readings, it is not clear what precisely triggered these changes. Is it the electrical field of the human, or is it the slight movement of the human body. It has been shown in previous works that plants are sensitive to both [3,24]. A second limitation is the small sample size. While our sample is quite small, we still find interesting and statistically significant results indicating a causal relationship between human body movements and electrical responses of the plant, thus motivating further in-depth research to better understand this fascinating phenomenon. We are currently repeating the experiment during an entire summer season, leading to two to four cohorts of test probands (lettuce, tomato plants, basil) and a much larger sample size.

## 8. Future Work

Future work in this field of study could focus on identifying more differences in plant responses to external stimuli. Within the domain of eurythmy, an intriguing avenue for experimentation would be to perform diverse eurythmy movements and subsequently explore the possibility of distinguishing between them using machine learning techniques. Similarly, can the plant distinguish between different human actors? Future work should also focus on understanding the underlying biological reasoning for plants to react to this type of stimuli. An initial experiment in this regard might be performed using water plants, as the ground for the spikerbox in this case would just be the water. We have already conducted preliminary experiments with potted orchids, where we only discovered an effect when the pot was placed into another larger bowl half filled with water.

## 9. Conclusions

We began this work with the hypothesis that plants can react to external stimuli in the form of non-physical interactions. To test this hypothesis, we set up a controlled experiment using a sample of plants and an actor trained in eurythmic gestures. We recorded the electric potential changes of the plants while the participant performed the gestures in the vicinity of the plants, while the same spikerbox readings were simultaneously taken from untreated plants farther away. The specialized equipment used for recording the voltage changes of the plants was designed to accurately measure any changes in electrical activity that may occur in response to external stimuli.

The results of our study showed a difference in the electric potential variations of the plants when the actor performed eurythmic gestures in their vicinity, compared to when the plants were left untreated.

These findings suggest that eurythmic gestures can have a measurable effect on the electrical activity of plants, and may have implications for our understanding of the relationship between human movements and plant behavior. Further research is needed to explore the mechanisms underlying this phenomenon and to investigate the potential applications of this knowledge in areas such as agriculture and environmental monitoring.

## Figures and Tables

**Figure 1 sensors-23-06971-f001:**
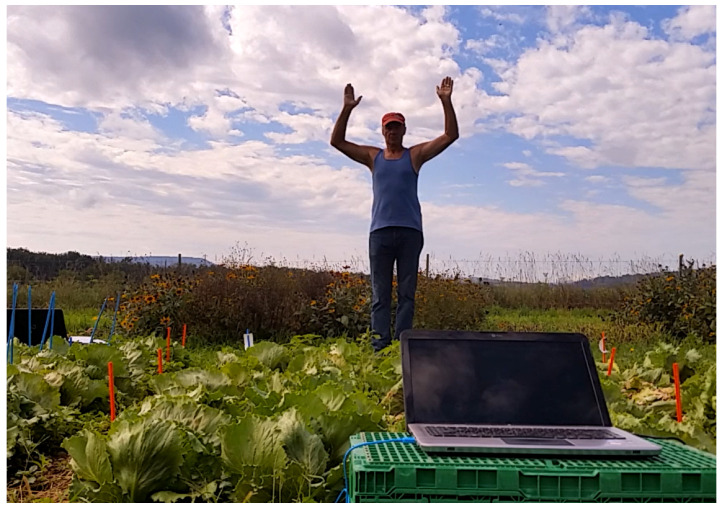
Image of eurythmic gestures being performed next to a field of lettuces.

**Figure 2 sensors-23-06971-f002:**
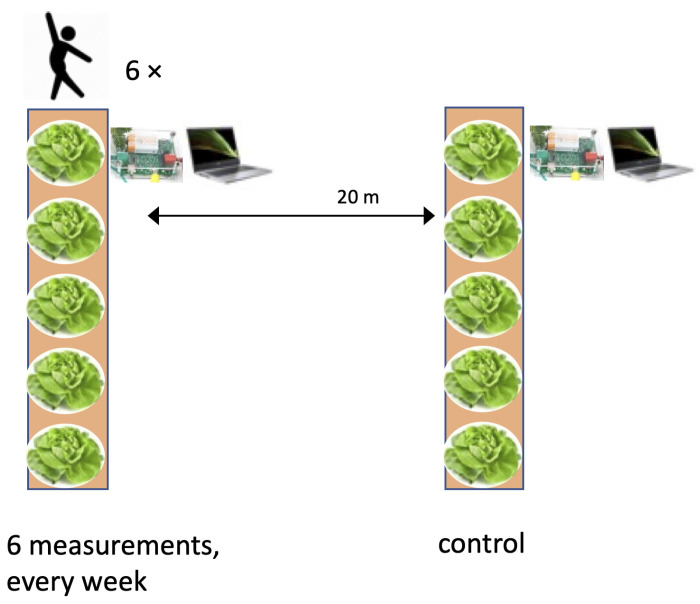
Experimental setup.

**Figure 3 sensors-23-06971-f003:**
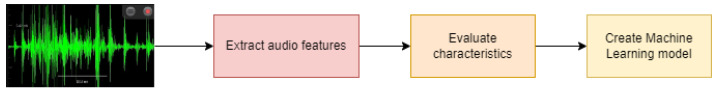
Method of experiment.

**Figure 4 sensors-23-06971-f004:**
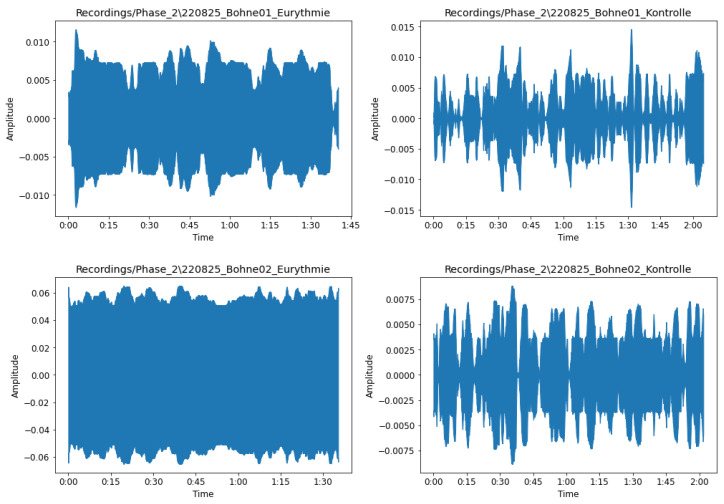
Diagram reflecting the potential variation in the plants for the untreated (**right**) plants and for the plants treated with eurythmy (**left**). Amplitude shows the relative voltage changes.

**Figure 5 sensors-23-06971-f005:**
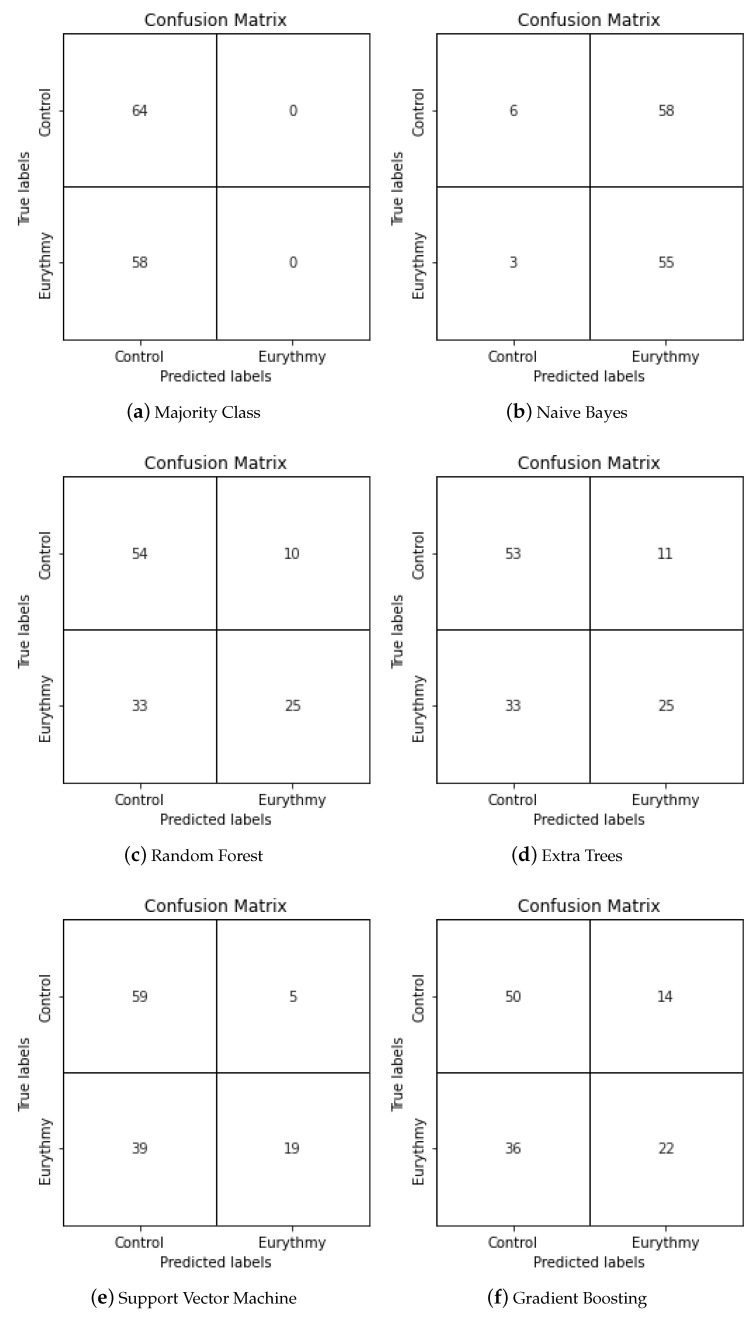
Confusion matrices of the different classical machine learning techniques applied to the same training and testing datasets.

**Figure 6 sensors-23-06971-f006:**
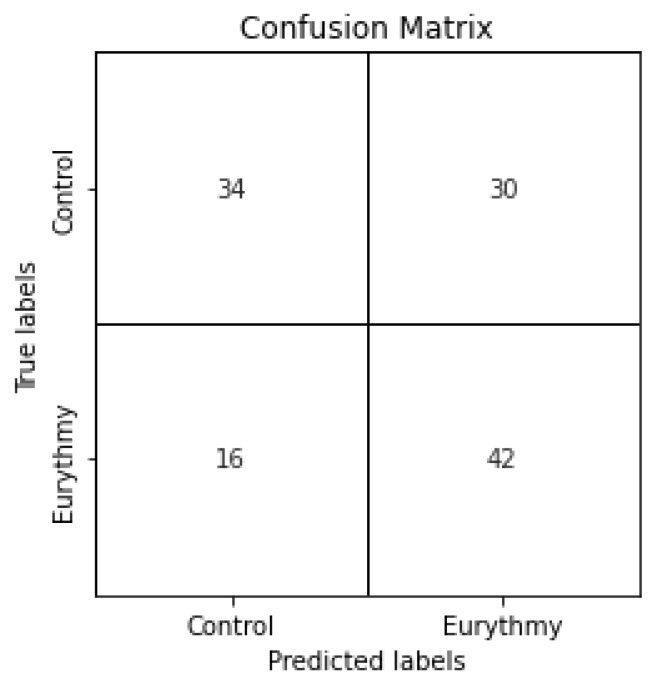
Confusion matrix for classification task with 100 epochs and a validation split of 0.2, selecting the weights with the best validation loss.

**Figure 7 sensors-23-06971-f007:**
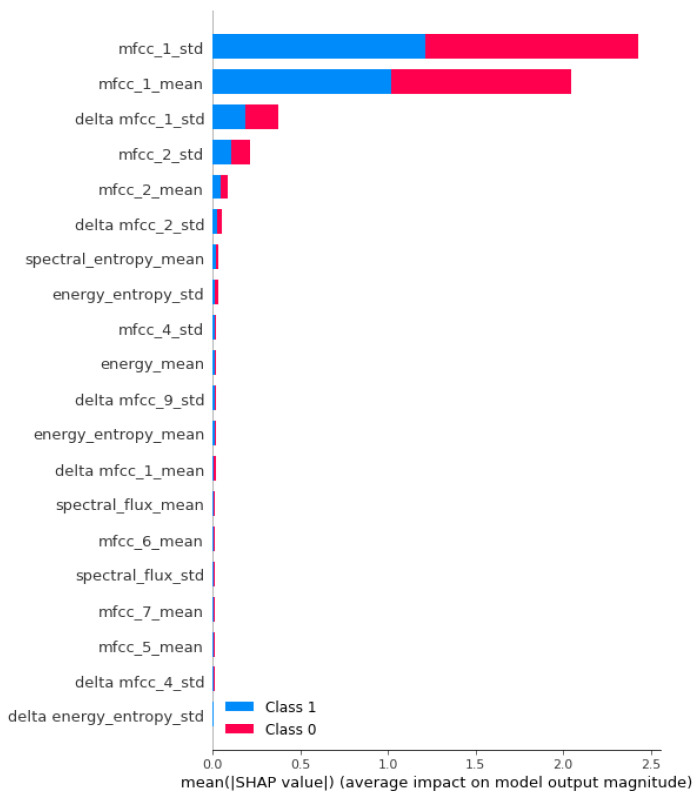
List of 20 most relevant SHAP values ordered from highest to lowest mean SHAP value.

**Figure 8 sensors-23-06971-f008:**
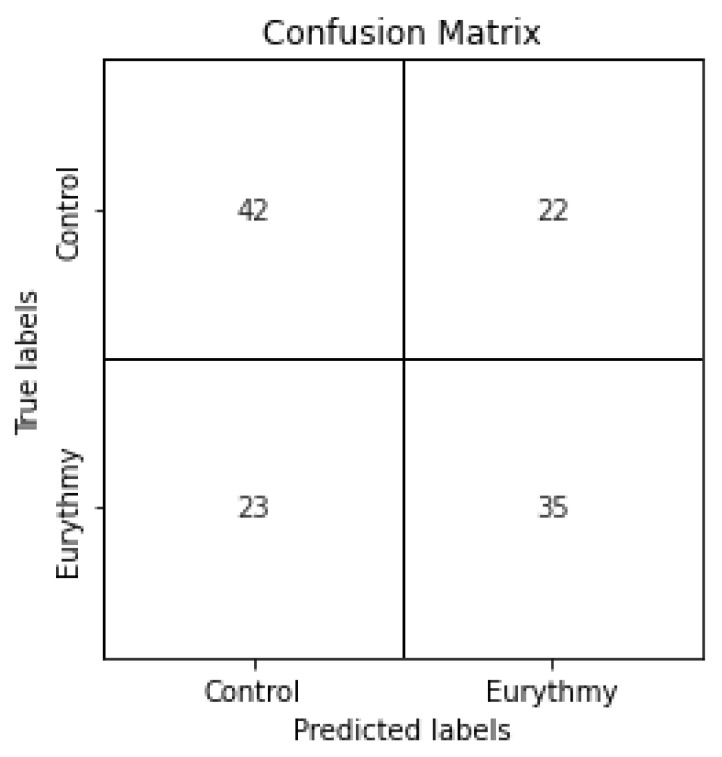
Confusion matrix for classification task with 200 epochs and a validation split of 0.25, selecting only the 10 features with the highest average SHAP values in the previous best model and the weights with the best validation loss.

**Table 1 sensors-23-06971-t001:** Description of the significant features computed with pyAudioAnalysis.

Feature	Description
delta spectral_spread_std	standard deviation of differences among the samples of the second central moment of the spectrum.
delta mfcc_[1…13]_std	standard deviation of differences among the samples of the Mel Frequency Cepstral Coefficients forming a cepstral representation where the frequency bands are not linear but distributed according to the mel-scale.
mfcc_[1…13]_std	standard deviation among the samples of the Mel Frequency Cepstral Coefficients forming a cepstral representation where the frequency bands are not linear but distributed according to the mel-scale.
delta spectral_centroid_std	standard deviation of differences among the samples of the center of gravity of the spectrum.
spectral_spread_std	standard deviation among the samples of the second central moment of the spectrum.

**Table 2 sensors-23-06971-t002:** Table showing the 14 features with the smallest *p*-value according to the performed *t*-test.

Feature	*p*-Value	Mean Eurythmy	Mean Control	Cohen’s d
delta spectral_spread_std	0.000004	0.058	0.049	−0.381
delta mfcc_1_std	0.000005	27.741	24.001	−0.378
delta mfcc_2_std	0.000011	0.968	0.854	−0.365
delta mfcc_4_std	0.000033	0.220	0.204	−0.344
delta mfcc_3_std	0.000092	0.185	0.173	−0.324
delta mfcc_12_std	0.000190	0.129	0.121	−0.309
delta spectral_centroid_std	0.000248	0.027	0.023	−0.303
delta mfcc_5_std	0.000270	0.147	0.140	−0.302
delta mfcc_6_std	0.000284	0.153	0.145	−0.300
mfcc_12_std	0.000357	0.090	0.084	−0.295
mfcc_10_std	0.000637	0.101	0.096	−0.283
delta mfcc_13_std	0.000948	0.117	0.110	−0.273
spectral_spread_std	0.000965	0.052	0.048	−0.273
delta mfcc_10_std	0.001002	0.145	0.137	−0.272

**Table 3 sensors-23-06971-t003:** Table showing the results of classical machine learning models. The best value for each metric has been highlighted.

Method	F1-Score	Accuracy	Precision	Recall
Majority class	0.0	0.52459	0.0	0.0
Naive Bayes	**0.64327**	0.5	0.48673	**0.94828**
Random Forest	0.53763	**0.64754**	0.71429	0.43103
Extra Trees	0.53191	0.63934	0.69444	0.43103
SVM	0.46341	0.63934	**0.79167**	0.32759
Gradient Boosting	0.46806	0.59016	0.61111	0.37931

## Data Availability

Not applicable.

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
