# Peer review of "Can Plants Sense Humans? Using Plants as Biosensors to Detect the Presence of Eurythmic Gestures"

_sensors, 2023, doi:10.3390/s23156971_

Round 1

Reviewer 1 Report

This paper proposed a method using plants as biosensors to detect the presence of eurythmic gestures.

The topic is interesting and ideas is novel. But there have some flaws as mentioned below.

 1. The paper is not well organized. For example, section 4, hypothesis.

2. The method of the paper has little innovation and too simple. As in Fig2, create machine learning model, what's the model, it's better for the author to well explain it.

3. The sample data is in a small size, it seems that could not prove your hypothesis.

4. The tester is perform eurythmic gestures next to the field of lettuce, did the authors consider the distance between the tester and the field. Is the distance a factor to the results.

5. The experiment was deployed in an agricultureal field, also the authors set one field as "control" group, but different field has different kinds of soil, and also could affect the results. The reviewer suggest the authors to do the experiment with water culture plants.

Minor editing of English language required.

Reviewer 2 Report

This paper presents our preliminary findings on the impact of human body movements on plants, specifically examining the effects of eurythmic gestures performed by human actors on lettuce and beans. While the topic is interesting, I acknowledge that there are several shortcomings in this research paper that need to be addressed.

1. The Abstract should clearly state the main objectives and scope of the investigation. It should also provide a description of the methods employed in the study.

2. The Introduction needs to be expanded to include a brief review of the relevant literature, highlighting existing problems. Additionally, it should state the specific method used in the investigation and provide reasons for its selection.

3. It is important to address other relevant literature, such as "The Visual Microphone- Passive Recovery of Sound from Video," to provide a comprehensive context for this study.

4. The Method section should provide sufficient details so that readers can replicate the experiments. The current version only offers a brief overview, and I believe that, considering the nature and scope of the investigation (i.e., sound recovery from vision), more specific details should be included.

5. The title should be positioned above the tables to follow standard formatting guidelines.

6. The hypothesis of this work is that recording voltage differences on different plants using specialized equipment enables the distinction of whether eurythmic gestures are being performed near the plant. However, the application value of this distinction should be further explained and discussed.

7. The experimental results should be used to present principles, relationships, and generalizations. Additionally, theoretical implications of this work should be discussed to provide a deeper understanding of its significance.

Overall, the English expression is good, but it still requires careful checking.

Reviewer 3 Report

Please address the following:

1. The main issue is given the small data size, is the deep neural network best choice for a machine learning model? The authors need to provide comparisons with other shallow models such as Random Forest, Support vector machine (SVM)

2. The authors present SHAP values to understand the feature importance for prediction. Following this analysis, it will be useful to perform a variable selection approach to find out if better classification performance can be achieved using a smaller set of important features.

3. The authors referred to a fully connected Deep Neural network as a 'Sequential Neural Network'. However, this term may be confused with sequential models such as recurrent neural network (RNN). Therefore, please rephrase. 

Round 2

Reviewer 1 Report

This paper is very interesting.

The authors has already answered the reviewer's comments.

It could be accepted in this version.

Reviewer 3 Report

Thanks for addressing the concerns and questions.